# The NF-κB regulator Bcl-3 restricts terminal differentiation and promotes memory cell formation of CD8+ T cells during viral infection

Hemant Jaiswal[1]*, Thomas Ciucci[2], Hongshan Wang[1], Wanhu Tang[1], Estefania Claudio[1], Philip M. Murphy[3], Rémy Bosselut[2], Ulrich Siebenlist[1]†

**1** Immune Activation Section, Laboratory of Molecular Immunology, National Institute of Allergy and Infectious Diseases, National Institutes of Health, Bethesda, Maryland, United States of America, **2** Laboratory of Immune Cell Biology, Center for Cancer Research, National Cancer Institute, National Institutes of Health, Bethesda, Maryland, United States of America, **3** Molecular Signaling Section, Laboratory of Molecular Immunology, National Institute of Allergy and Infectious Diseases, National Institutes of Health, Bethesda, Maryland, United States of America

† Deceased.
* hemant.jaiswal@nih.gov

**Data Availability Statement:** The authors confirm that all data underlying the findings are fully available without restriction. Sequence data are

## Abstract

Bcl-3 is an atypical member of the IκB family that acts in the nucleus to modulate transcription of many NF-κB targets in a highly context-dependent manner. Accordingly, complete *Bcl-3*$^{-/-}$ mice have diverse defects in both innate and adaptive immune responses; however, direct effects of Bcl-3 action in individual immune cell types have not been clearly defined. Here, we document a cell-autonomous role for Bcl-3 in CD8+ T cell differentiation during the response to lymphocytic choriomeningitis virus infection. Single-cell RNA-seq and flow cytometric analysis of virus-specific *Bcl3*$^{-/-}$ CD8+ T cells revealed that differentiation was skewed towards terminal effector cells at the expense of memory precursor effector cells (MPECs). Accordingly, *Bcl3*$^{-/-}$ CD8+ T cells exhibited reduced memory cell formation and a defective recall response. Conversely, Bcl-3-overexpression in transgenic CD8+ T cells enhanced MPEC formation but reduced effector cell differentiation. Together, our results establish Bcl-3 as an autonomous determinant of memory/terminal effector cell balance during CD8+ T cell differentiation in response to acute viral infection. Our results provide proof-of-principle for targeting Bcl-3 pharmacologically to optimize adaptive immune responses to infectious agents, cancer cells, vaccines and other stimuli that induce CD8+ T cell differentiation.

## Author summary

Effective immunity to previously encountered pathogens or vaccines depends on efficient formation of memory T and B cells. A major cornerstone is the generation of memory CD8+ T cells, which kill cells infected with intracellular pathogens, including viruses. In addition, memory CD8+ T cells are critical for adoptive immunotherapy to eliminate targeted tumor cells. Hence, it is essential to have a detailed understanding of pathways and molecular factors that regulate T cell differentiation/memory formation during the initial

deposited in the NCBI Gene Expression Omnibus under the accession number GSE148660.

**Funding:** This project was supported by the Intramural Research Program of National Institute of Allergy and Infectious Diseases (NIAID), NIH awarded to US and the Intramural Research Program of the National Cancer Institute, Center for Cancer Research, NIH awarded to RB. Support from CCR Single Cell Analysis Facility and CCR Sequencing Facility was funded by FNLCR Contract HHSN261200800001E. This work utilized the computational resources of the NIH HPC Biowulf cluster (http://hpc.nih.gov). The funders had no role in study design, data collection and analysis, decision to publish, or preparation of the manuscript.

**Competing interests:** The authors have declared that no competing interests exist. Author Ulrich Siebenlist was unable to confirm their authorship contributions. On their behalf, the corresponding author has reported their contributions to the best of their knowledge.

immune effector response. Here we show a critical cell-autonomous role for Bcl-3, an atypical member of the IκB family, in the regulation of CD8$^+$ T cell differentiation/memory formation during acute viral infection with lymphocytic choriomeningitis virus. We found that Bcl-3 dampens the terminal differentiation of CD8$^+$ T cells, which normally results in death at the end of the primary effector response, and instead, promotes the formation of memory precursor cells early on. This led to a robust memory response and much improved viral clearance upon re-challenge. Our results provide new insight into the molecular regulation of CD8$^+$ T cell effector response and memory cell generation, establishing Bcl-3 and its pathways as a target to optimize CD8$^+$ T cell effector responses to various immunogens including infectious agents, cancer cells, and vaccines.

## Introduction

Memory CD8$^+$ T cell generation is a hallmark of adaptive immune responses. These memory CD8$^+$ T cells possess a blueprint of past infection that directs rapid and effective responses to repeat infection with the same pathogen. Interactions of naïve CD8$^+$ T cells with antigen presenting cells in an infectious environment leads to very early asymmetric divisions and massive expansions generating a heterogeneous pool of effector CD8$^+$ T cells [1,2]. Short lived effector cells (SLECs) identified as KLRG1$^{hi}$ IL-7Rα$^{lo}$ cells form the major fraction of CD8$^+$ T cell effector responses. Upon further maturation, SLECs show profound cytotoxic activity during the primary response, but then undergo massive apoptosis during the contraction phase as the infection resolves. However, one minor subset of activated CD8$^+$ T cells known as memory precursor effector cells (MPECs; KLRG1$^{lo}$ IL-7Rα$^{hi}$) largely survive the contraction and form long-lived memory CD8$^+$ T cells [3]. Extensive research in the field has led to the identification of various cellular interactions, cytokines and transcription factors involved in generating heterogeneous populations of CD8$^+$ T cells during differentiation. In particular, inflammatory antimicrobial cytokines such as IL-12 [4], type I interferons [5] and IFN-γ [6] are major drivers of CD8$^+$ T cell terminal differentiation. Together, the strength and duration of inflammatory cytokine and antigen stimulation induce a graded expression of a set of transcription factors (TFs) including T-bet [3], *Id2* [7], Blimp1 [8] and *Zeb2* [9] that promote terminal differentiation, in part by supporting effector gene expression. However, at the same time and also dependent on antigen strength and duration of stimulation, TFs like Eomes [10], *Bcl6* [11], *Id3* [12], *Foxo1* [13] and Tcf1 [14,15] counterbalance TFs that support effector differentiation to maintain naïve-like characteristics in activated CD8$^+$ T cells. Even in the case of persistent infection, a Tcf1$^{hi}$ subset of effector CD8$^+$ T cells has been shown to have naïve or progenitor-like characteristics. These cells maintain their stemness and continuously feed the effector pool to fight against persistent pathogens making them critical for chronic infections and anti-tumor responses [16,17]. However, how this balance of memory and effector cells is determined has not been fully delineated.

Bcl-3 (B cell lymphoma factor 3) is an atypical member of the IκB family. Unlike classical IκBs, Bcl-3 is not degraded in response to activation signals, but instead is often induced at the mRNA and protein levels, whereupon it acts in the nucleus to modulate transcription of NF-κB target genes by associating with p50/NF-κB1 and p52/NF-κB2 homodimers on DNA [18–21]. The p50 and p52 homodimer complexes do not trans-activate by themselves, but association with Bcl-3 may promote or inhibit transcription of targets in a highly gene- and context-dependent manner, possibly influenced by post-translational modifications [22,23]. Bcl-3 has profound immunoregulatory effects, including modulation of central tolerance, secondary

lymphoid organ development [24] and B cell development [25]. Thus, complete *Bcl3*[-/-] mice exhibit notably impaired innate and adaptive immune responses, including impaired germinal center formation [25] and are highly susceptible to *Klebsiella pneumoniae* [26] and *Toxoplasma gondii* infections [27].

Regarding autonomous roles in T cells, we have shown that Bcl-3 regulates plasticity and pathogenicity of autoimmune CD4[+] T cells in the T cell transfer-induced model of colitis as well as in experimental autoimmune encephalomyelitis (EAE), where Bcl-3 deficient CD4[+] T cells attain a less pathogenic IL-17-producing Th17-like phenotype, instead of the pathogenic, IFN-γ producing Th1-like (ex-Th17) phenotype observed in Bcl-3 sufficient mice [28]. Also, adjuvant-induced survival of both CD4[+] and CD8[+] T cells was found to correlate with Bcl-3 expression in an *in vivo* model of superantigen stimulation [29]. Subsequently, however Bcl-3 was shown not to be necessary for T cell survival in this context. Instead, Bcl-3 was reported to aid the production of IFN-γ by T cells in this superantigen model as determined by T cell re-stimulation *ex vivo* [30]. On the other hand, co-stimulation of Bcl-3-sufficient but not Bcl-3-insufficient T cells with LPS or IL-12 did aid in the survival of T cells *in vitro*, including CD8[+] T cells [31,32]. Furthermore, overexpression of a Bcl-3 transgene in T cells *in vivo* or *in vitro* promoted survival and slowed early T cell activation/proliferation in the superantigen stimulation model [33]. Despite these advances in knowledge, the role played by Bcl-3 during one of the most important biologic functions of T cells, control of viral infection, has not been previously investigated. To begin to address this gap, here, we studied the role of Bcl-3 in CD8[+] T cell differentiation during acute lymphocytic choriomeningitis virus (LCMV) infection. An in-depth investigation of the CD8[+] T cell primary response by single-cell RNA-sequencing (scRNA-seq) revealed that Bcl-3 deficiency affected the CD8[+] T cells differentiation program, favoring terminal effector cells at the expense of MPECs. Accordingly, *Bcl3*[-/-] CD8[+] T cells exhibited reduced memory T cell formation as well as defective recall responses upon LCMV re-challenge. Furthermore, the primary response of Bcl-3-overexpressing transgenic mice suggested that Bcl-3 expression in CD8[+] T cells is critical for MPEC formation. The presence of the transgene first dampened CD8[+] T cell activation/proliferation and this in turn appeared to limit terminal differentiation of these cells and instead supported generation of MPECs. Our study indicates that Bcl-3 is a critical factor in CD8[+] T cell differentiation, promoting the maintenance of their naïve/progenitor-like characteristics and thus allowing them to survive better and to more efficiently form MPECs and eventually memory cells.

## Results

### The quantitative response of CD8[+] T cells to acute LCMV infection is independent of Bcl-3 in these cells

Our prior work demonstrated critical cell-autonomous contributions of Bcl-3 in maintaining pathogenic Th1-like, IFN-γ producing CD4[+] T cells in the context of EAE and colitis [28]. Other reports–albeit with partly divergent conclusions–additionally suggested roles for Bcl-3 in either survival of CD8[+] T cells *in vitro* or their IFN-γ production *ex vivo* [29–33]. These findings prompted us to systematically investigate the function of Bcl-3 in CD8[+] T cells *in vivo*. The LCMV acute infection model is a well-studied *in vivo* model for acute responses and differentiation of CD8[+] T cells. We first confirmed that *Bcl3* mRNA was transiently induced in CD8[+] T cells upon CD3 plus CD28 co-stimulation; induction was apparent by 3 hours and peaked at 6 hours, eventually falling to pre-stimulation levels by 24 h (S1A Fig). We also detected *Bcl3* mRNA induction in CD8[+] T cells from the spleen of LCMV Armstrong-infected mice (S1B Fig).

Infection with the LCMV Armstrong strain is known to result in a robust T cell response with clearance of the virus by day 8–10 post infection (p.i.) [34]. To specifically check the cell-autonomous role of Bcl-3 in CD8$^+$ T cells in the anti-LCMV response, we generated mixed bone marrow chimeric mice by transferring congenic wild-type (WT, CD45.1) and *Bcl3$^{-/-}$* (Bcl-3 knockout, CD45.2) bone marrow cells into sub-lethally irradiated *Rag1$^{-/-}$* mice. After allowing at least 6 weeks for reconstitution, chimeric mice were infected with LCMV Armstrong and CD8$^+$ T cell responses were compared 8 days p.i. (Fig 1A–1D). Spleens and mesenteric lymph nodes (mLNs) from infected chimeric mice showed no difference between WT and Bcl-3 KO CD8$^+$ T cells in their proportion of LCMV-specific cells (H-2D$^b$-GP33$^+$ CD8$^+$) (Fig 1A and 1B). Moreover, intracellular cytokine staining of splenocytes stimulated *ex vivo* with GP33 peptides revealed no difference between WT and *Bcl3$^{-/-}$* donor cells in the percentages of both IFN-γ single positive and IFN-γ + TNF-α double positive LCMV-specific CD8$^+$ T cells (Fig 1C and 1D).

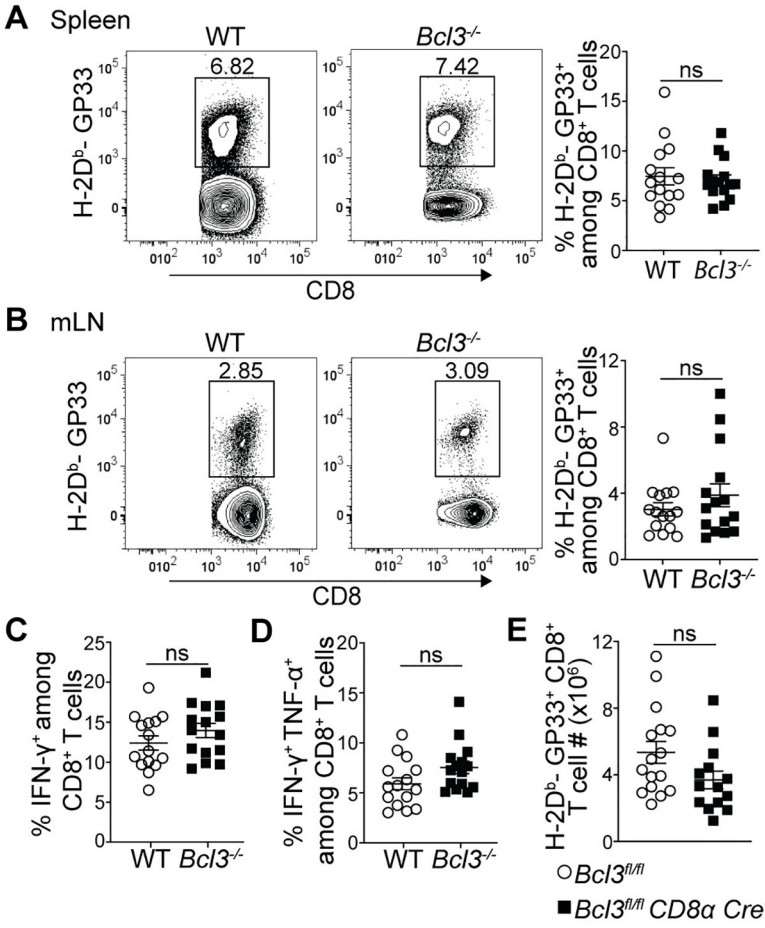

**Fig 1. Bcl-3 deficient and wild-type CD8$^+$ T cells show comparable overall anti-viral responses against acute LCMV infection.** (**A-D**) Mixed bone marrow chimeric mice were analyzed 8 days after LCMV Armstrong infection. Representative contour plots show proportions of H-2D$^b$-GP33 tetramer-specific cells among WT or Bcl-3 KO CD8$^+$ T cells in spleen (**A**; left panels) and mesenteric lymph node (mLN) (**B**; left panels). Right graphs summarize data from three independent experiments with n = 15 mice for each CD8$^+$ T cell genotype. Proportions of IFN-γ (**C**) and IFN-γ + TNF-α (**D**) secreting WT or *Bcl3$^{-/-}$* CD8$^+$ T cells upon GP33 peptide stimulation of splenocytes *ex vivo*. Graphs summarize data from three independent experiments with n = 15 mice for each CD8$^+$ T cell genotype. (**E**) Absolute number of H-2D$^b$-GP33$^+$ CD8$^+$ T cells in spleens of *Bcl3$^{fl/fl}$ CD8α Cre* or *Bcl3$^{fl/fl}$* (control) mice. Data are averages of three independent experiments with n = 14–16 mice for each genotype. Error bars indicate standard error of the mean (SEM). ns = not significant.

Although reconstitution of total WT and KO cells in bone marrow chimeric mice was close to 50:50 as confirmed in blood after 6 weeks of transfer, the proportion of Bcl3-deficient CD8[+] T cells was reduced compared to WT cells, resulting in 1.4-fold higher CD4/CD8 ratios for KO cells (S1C Fig). This suggested a possible competitive advantage of WT CD8[+] T cells over Bcl-3-deficient cells or of Bcl-3-deficient CD4[+] T cells over WT cells (or both) and thus prevented a comparison of absolute numbers of antigen-specific cells from chimeric mice in response to LCMV infection. To address this, we also generated CD8[+] T cell-specific conditional KO mice by crossing *Bcl3^(fl/fl)* mice with *CD8α-Cre* mice. In *CD8α-Cre* mice, the E8I enhancer region drives Cre expression specifically in peripheral mature CD8[+] T cells [35,36], which led to robust deletion of Bcl-3 in these cells (S1D Fig). Consistent with the post-thymic expression of Cre, there was no evidence of impaired CD8[+] T cell development and these conditional knock-out mice had the same number of naïve (CD62L^(hi) CD44^(lo)) spleen CD8[+] T cells as control mice (S1E Fig). Comparison of absolute numbers of GP33 epitope-specific CD8[+] T cells on day 8 p. i. in these mice also ruled out any obvious defect in generating anti-LCMV-specific CD8[+] T cells due to loss of Bcl-3 (Fig 1E).

## Bcl-3 deficiency diverts the CD8[+] T cell differentiation program to terminal effectors at the expense of MPECs

While Bcl-3 deficiency did not lead to any obvious difference in terms of generating LCMV epitope-specific CD8[+] T cells after acute infection, the flow cytometric analysis did not rule out possible differences in subsets generated during CD8[+] T cell differentiation. We therefore investigated cell-autonomous effects of Bcl-3 on CD8[+] T cell differentiation using unbiased transcriptomic analysis to define its impact on subsets. To do so, mixed bone marrow chimeras generated with WT and *Bcl3^(-/-)* bone marrow were infected with LCMV Armstrong and virus-specific CD8[+] T cells (H-2D^b-GP33[+] CD8[+]) were analyzed 8 days p.i. by single-cell RNA-sequencing (scRNA-seq) (S2A Fig). Splenic WT and *Bcl3^(-/-)* GP33 antigen-specific CD8[+] T cells from the same chimeric animal were sorted based on CD45 allelic markers and captured using a droplet-based 10x Genomics Chromium platform [37]. After quality controls, filtering and dimensional reduction of the data, both WT and *Bcl3^(-/-)* cells were merged together. Dimensional reduction of the data and its visualization using uniform manifold approximation and projection (UMAP) showed that both populations largely overlapped, suggesting a limited impact of Bcl-3 deletion on the overall transcriptional landscape of virus-specific CD8[+] T cells (Fig 2A).

To gain further insights, we performed unsupervised clustering of the combined cell population, resulting in 7 clusters including cells of both genotypes. We then analyzed gene expression across clusters of WT or *Bcl3^(-/-)* cells (Fig 2B). In the WT population, we detected 2 distinct groups of cells. First, a large group of effector cells (Clusters A-F, 93%) expressed high levels of canonical effector genes including *Gzmb*, *Ifng* and *Prf1* (encoding granzyme B, IFN-γ and perforin, respectively). In addition to the large cluster A (52%), we identified smaller effector clusters, including cluster B (13%), which appears to be closely related to A, as well as a cluster of terminally differentiated effector cells (Cluster C, 12%) expressing *Zeb2* and *Cx3cr1*, two clusters of cycling cells expressing cell-cycle associated genes (Clusters D and E, 8 and 5%, respectively) and a cluster of interferon-stimulated cells expressing *Irf7* and other interferon-sensitive genes (Cluster F, 3%). Additionally, unsupervised clustering distinguished a small cluster of memory precursor effector cells (MPECs; Cluster G, 6%) that does not express effector genes; instead, this cluster contains cells expressing high levels of memory associated genes like *Il7r*, *Tcf7* and *Ccr7* (encoding IL-7R, TCF-1 and CCR7, respectively). Interestingly, in the *Bcl3^(-/-)* population, although the pattern of expression and cluster types appeared largely similar

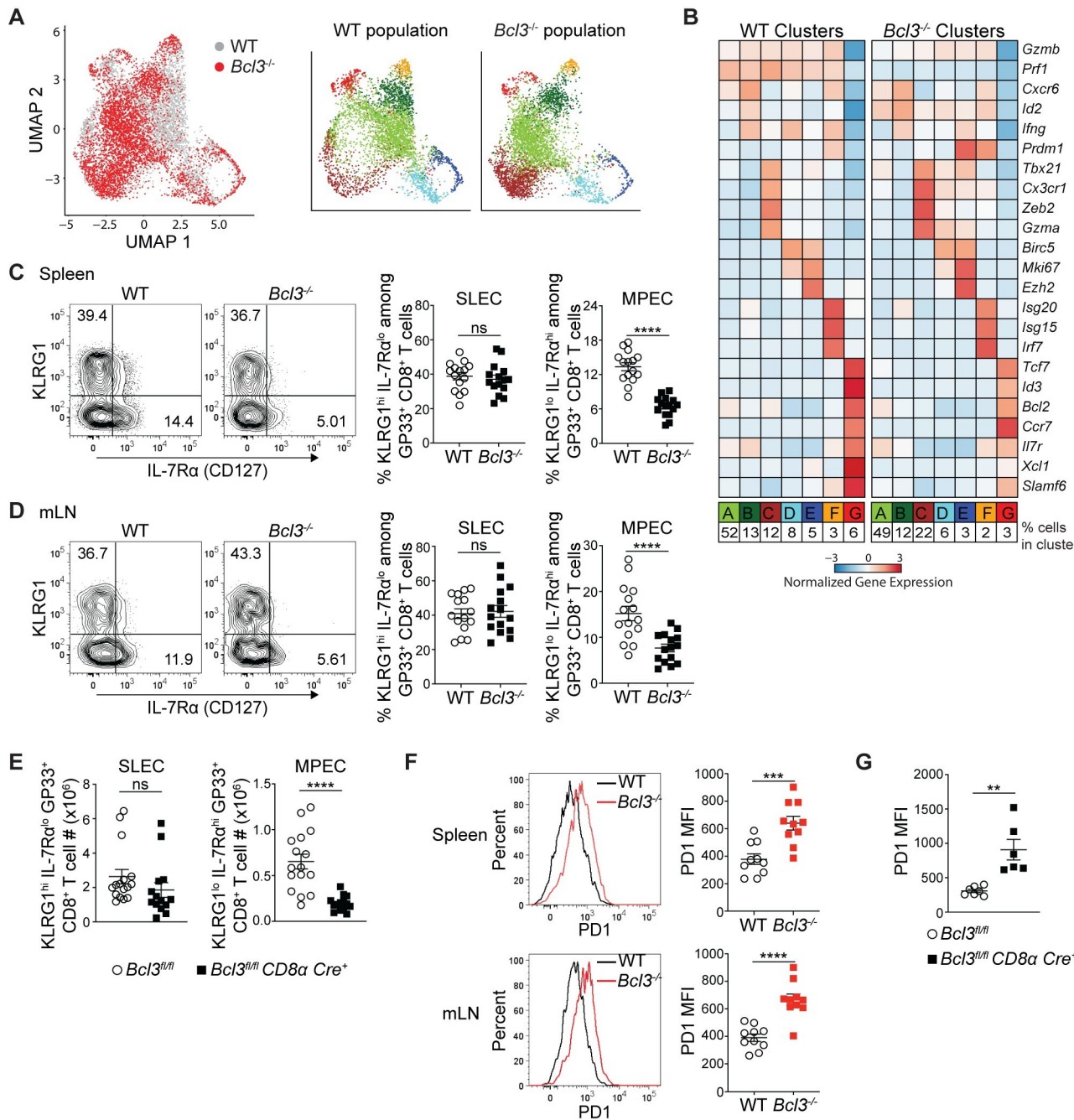

**Fig 2. Bcl-3 supports formation of memory precursor effector CD8+ T cells.** (**A** and **B**) Bcl-3 KO and WT H-2D^b-GP33+ cells isolated on d 8 p.i. from mixed bone marrow chimeras were analyzed by scRNA-seq. (**A**) Left UMAP plot shows Bcl-3 KO and WT cells colored by genotype as indicated. Bcl-3 KO and WT populations were subsequently clustered independently and detected clusters were projected on the previous UMAP plot (right panels) and color-coded. (**B**) Heatmap shows row-standardized expression of selected genes among clusters of Bcl-3 KO and WT cells displayed in (A). For each genotype, percentages listed at bottom indicate the size of individual clusters relative to the total population. (**C** and **D**) 8 d p.i. mixed bone marrow chimeras were analyzed for SLECs (short lived effector cells, KLRG1^hi IL-7Rα^lo) and MPECs (memory precursor effector cells, KLRG1^lo IL-7Rα^hi) subsets. Representative contour plots show the proportions of SLEC and MPEC cells in WT or *Bcl3^-/-* H-D^b-GP33+ CD8+ T cells in spleen (**C**; left panels) and mLN (**D**; left panels). Right graphs summarize data from three independent experiments with n = 15 mice for each CD8+ T cell genotype. (**E**) Absolute numbers of SLEC and MPEC subsets in spleen of *Bcl3^fl/fl CD8α Cre* or control mice 8 days after LCMV Armstrong infection. Data are averages of three independent experiments with n = 14–16 mice for each genotype. (**F**) Representative histograms show the intensity of PD1 expression on WT or *Bcl3^-/-* H-D^b-GP33+ CD8+ T cells from spleen and mLN of mixed bone marrow chimera 8 d p.i. with LCMV Armstrong (left panel). Right graphs summarize data from two independent experiments with n = 10 mice for each CD8+ T cell genotype. (**G**) Summary of PD1 expression on LCMV antigen GP33 specific CD8+ T cells in spleen. Data are representative of two independent experiments with n = 5 mice in each group. Error bars indicate SEM. **p<0.01, ***p<0.001, ****p<0.0001. ns = not significant.

to those in WT cells, we observed a reduction in the MPEC fraction (Cluster G—6% in WT vs 3% in *Bcl3*$^{-/-}$) and an increase in the proportion of terminally differentiated effectors (Cluster C—12% in WT vs 22% in *Bcl3*$^{-/-}$). These results were confirmed in a second biologically independent replicate (S2B and S2C Fig).

To validate the scRNA-seq findings, we analyzed markers of effector and memory precursor subsets in mice after acute infection by flow cytometry. Consistent with the scRNA-seq pattern, Bcl-3-deficient CD8$^+$ T cells showed reduced proportions compared to WT of the MPEC subset (KLRG1$^{lo}$ IL-7Rα$^{hi}$ [CD127]) among GP33$^+$ CD8$^+$ T cells in both spleen and mLN from mixed bone marrow chimeras (Fig 2C and 2D). Analysis of spleen from *Bcl3*$^{fl/fl}$ *CD8α Cre*$^+$ mice 8 d p.i. also showed a reduced proportion (S3A and S3B Fig) as well as reduced absolute numbers of GP33$^+$-specific KLRG1$^{lo}$ IL-7Rα$^{hi}$ CD8$^+$ MPECs compared to controls (Fig 2E). Analysis of SLEC and MPEC subsets from WT mice suggested low level *Bcl3* expression in these subsets (S3C Fig). However, surface staining with these markers did not show upregulation of SLECs (KLRG1$^{hi}$ IL-7Rα$^{lo}$ subset) in either the chimeric mice or the CD8$^+$ T cell conditional KO mice (Fig 2C–2E). This likely reflects the fact that the KLRG1$^{hi}$ IL-7Rα$^{lo}$ subset includes all effectors and not just the smaller population of terminal effector cells, which were the ones notably upregulated in *Bcl3*$^{-/-}$ mice by scRNA-seq. However, we did observe higher PD1 expression by antigen-specific Bcl-3-deficient CD8$^+$ T cells in chimeric mice (Fig 2F). Similarly, higher expression of PD1 was also noted on antigen-specific CD8$^+$ T cells from conditional *CD8α Cre*$^+$ KO mice (Fig 2G). Other effector markers like perforin, granzyme B and T-bet were comparable between control and conditional *CD8α Cre*$^+$ KO mice (S3D–S3F Fig). However, consistent with the role of Eomes in memory CD8$^+$ T cell development [10], GP33-specific CD8$^+$ T cells from *Bcl3*$^{fl/fl}$ *CD8α Cre*$^+$ showed less expression of Eomes compared their control counterparts (S3G Fig). Furthermore, the conditional knockout mice contained higher serum IFN-γ levels compared to WT controls 5 d p.i. (S3H Fig). Higher PD1 expression and enhanced cytokine production suggest that Bcl-3-deficient CD8$^+$ T cells were more differentiated compared to WT cells at this stage. Altogether these findings document a cell-autonomous regulatory role of Bcl-3 in formation of MPECs, most likely by restricting CD8$^+$ T cell differentiation towards terminal effectors. To examine if this was caused by defective proliferation or survival of Bcl-3 deficient CD8$^+$ T cells, we analyzed the proliferation and survival of these cells *in vitro* and *in vivo*. Activation of purified CD8$^+$ T cells from *Bcl3*$^{-/-}$ and control mice with CD3 and CD28 resulted in comparable proliferation after 3 days of culture, as indicated by dilution of the proliferation dye cell trace violet (S4A and S4B Fig) and by cell counts (S4C Fig). In these, experiments, CD8$^+$ T cells from both genotypes also showed comparable apoptosis as checked by annexin V staining (S4D Fig). *In vivo* analyses found no detectable difference between *Bcl-3* null and control CD8$^+$ T cells for proliferation and apoptotic cell death upon LCMV infection (S4E–S4H Fig).

## Efficient formation of memory CD8$^+$ T cells is Bcl-3 dependent

MPECs have been shown to survive the contraction phase much better than SLECs and to form long lasting CD8$^+$ T cell memory. Thus, any defect in MPEC formation during the acute effector response might lead to altered memory CD8$^+$ T cell formation. Hence, we checked the status of LCMV epitope-specific CD8$^+$ T cells at least 8 weeks after acute infection of mixed bone marrow chimeric mice, as well as of *Bcl3*$^{fl/fl}$ *CD8α Cre*$^+$ mice (Fig 3). MHC-I tetramer staining after 8 weeks p.i. of chimeric mice with LCMV Armstrong showed reduced proportions of GP33 epitope-specific CD8$^+$ T cells in the absence of Bcl-3 (Fig 3A). Intracellular cytokine staining for IFN-γ- and IFN-γ + TNF-α-producing CD8$^+$ T cells upon GP33 peptide stimulation also indicated defective memory formation by Bcl-3-deficient CD8$^+$ T cells against

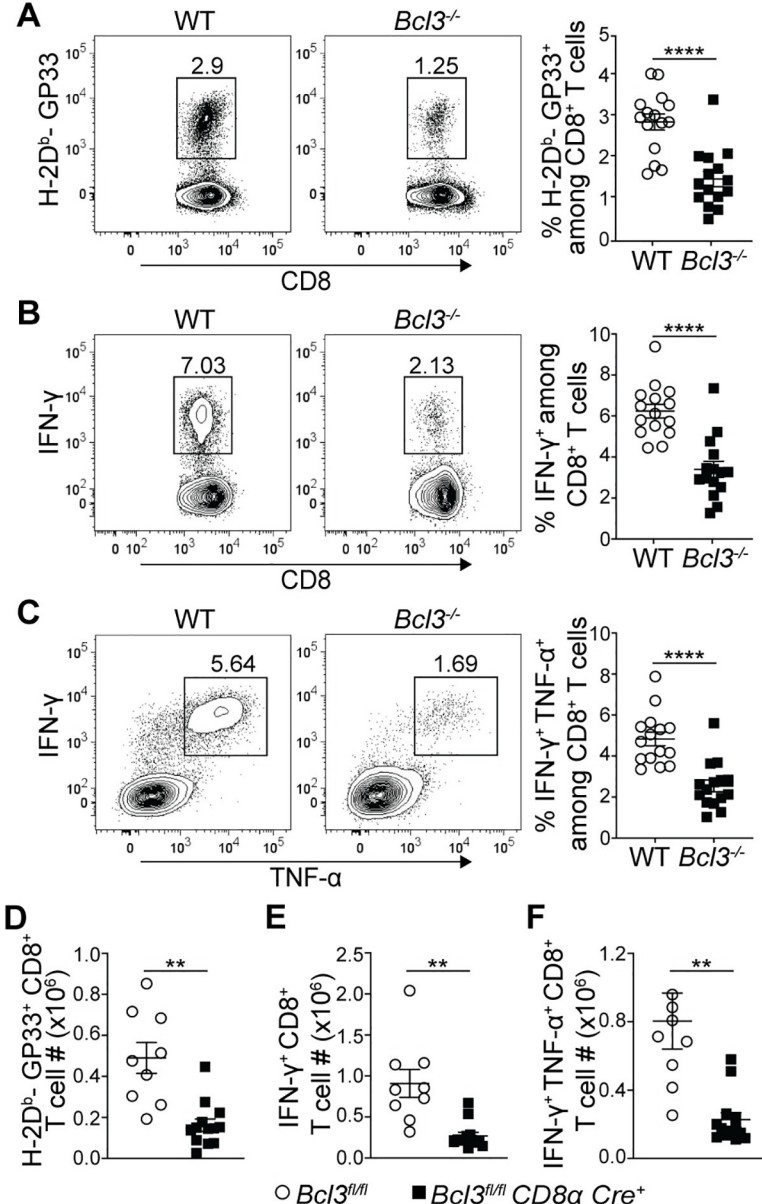

**Fig 3. Bcl-3 is essential for efficient memory CD8⁺ T cell generation.** (**A-C**) Spleens of mixed bone marrow chimeric mice were analyzed at least 8 weeks after LCMV Armstrong infection. (**A**) Representative contour plots show proportions of H-2D^b^-GP33 tetramer specific cells among WT or Bcl-3 KO CD8⁺ T cells in spleen (left). Right graph summarizes data from three independent experiments with n = 15 mice. Representative contour plots show proportions of IFN-γ (**B**; left panels) and IFN-γ + TNF-α (**C**; left panels) producing CD8⁺ T cells upon GP33 peptide stimulation *ex vivo* (left). Right graphs summarize data from three independent experiments with n = 15 mice for each CD8⁺ T cell genotype. (**D-F**) *Bcl3^fl/fl^* and *Bcl3^fl/fl^ CD8α Cre* mice were infected with LCMV Armstrong and spleens were analyzed at least 8 weeks post infection. Data are from 2 independent experiments with n = 9–13 mice for each genotype. (**D**) Absolute numbers of H-2D^b^-GP33⁺ CD8⁺ T cells in spleen. (**E and F**) Absolute numbers of IFN-γ positive (**E**) and IFN-γ + TNF-α positive (**F**) CD8⁺ T cells after GP33 peptide stimulation *ex vivo*. Error bars indicate SEM. \*\*p<0.01, \*\*\*\*p<0.0001.

LCMV (Fig 3B and 3C). In addition, analysis of CD8⁺ T cells in conditional *CD8α-Cre* knock-out mice after 8 weeks p.i. with LCMV revealed reduced absolute numbers of GP33⁺ CD8⁺ T cells compared to controls (Fig 3D). The conditional knock-out mice also showed reduced

numbers of IFN-γ- and IFN-γ + TNF-α-secreting cells upon *ex vivo* stimulation of splenocytes with GP33 peptides (Fig 3E and 3F). However, we did not observe any defect in commitment towards central memory cell ($T_{CM}$) formation by GP62-specific CD8$^+$ T cells in the absence of Bcl-3, as observed by comparable proportions of CD62L$^{hi}$ IL-7Rα$^{hi}$ $T_{CM}$ cells among WT and Bcl-3 knock-out cells in mixed bone marrow chimeric mice (S5 Fig). $T_{CM}$ cells represent a long-lasting small fraction of memory cells that preferentially recirculates from blood through secondary lymphoid organs. These data establish Bcl-3 as a crucial transcription factor for formation of overall CD8$^+$ T cell memory by promoting the formation of MPECs during the effector response.

## Bcl-3 deficient CD8$^+$ T cells show defective recall responses

Defects in memory T cell formation by Bcl-3-deficient CD8$^+$ T cells suggested a possible defect in recall responses. To investigate this, mixed bone marrow chimeric mice infected with LCMV Armstrong were re-challenged at least 8 weeks later with LCMV clone 13 to assess the overall CD8$^+$ T cell memory response. Bcl-3-deficient CD8$^+$ T cells showed reduced proportions of LCMV GP33 epitope-specific cells in spleen compared to their WT counterparts at day 5 post re-challenge (Fig 4A). This defective CD8$^+$ T cell response in the absence of Bcl-3 was confirmed by reduced proportions of IFN-γ single positive (Fig 4B) and IFN-γ + TNF-α double positive cells (Fig 4C) in Bcl-3-deficient compared to Bcl-3-sufficient CD8$^+$ T cells upon *ex vivo* stimulation with GP33 peptide.

Recall experiments with mice bearing conditional CD8$^+$ T cell-specific Bcl-3 knockouts confirmed the above findings, as we observed reduced absolute numbers of GP33$^+$ CD8$^+$ T cells directed against LCMV clone 13 compared to controls in this re-challenge scenario as well (Fig 4D). Furthermore, *ex vivo* stimulation with GP33 peptide resulted in reduced numbers of IFN-γ single positive (Fig 4E) as well as IFN-γ + TNF-α double positive CD8$^+$ T cells (Fig 4F). To further strengthen our observation, we measured the level of LCMV-specific transcripts in mouse serum 4 days after re-challenge with clone 13. The Bcl-3-deficient CD8$^+$ T cell-bearing mice showed compromised virus clearance compared to the control mice (Fig 4G). These results strongly suggest that Bcl-3 is an important transcriptional regulator to maintain proper recall responses of CD8$^+$ T cells against LCMV infection.

## Bcl-3 overexpression restricts terminal differentiation and supports MPEC formation

The above data suggested that Bcl-3 deficiency diverted CD8$^+$ T cell differentiation towards terminal effectors leading to decreased MPECs and thus poor memory formation and recall responses. Hence, we also checked the effect of Bcl-3 over-expression in CD8$^+$ T cells differentiation by using Bcl-3 transgenic mice. Bcl-3 transgenic (*Bcl3 Tg*) mice provide constitutive Bcl-3 expression in targeted cells where Cre mediated excision removes a stop codon upstream of the transgene to drive its expression [38]. We crossed *Bcl3 Tg* mice with *CD4-Cre* mice to facilitate Bcl-3 expression in all T cells. Acute LCMV infection of transgenic mice showed reduced proportions and absolute numbers of GP33$^+$ CD8$^+$ T cells in spleen of *CD4-Cre*$^+$ mice 8 d p.i. (Fig 5A). These findings were further supported by reduced proportions and absolute numbers of IFN-γ- or INF-γ + TNF-α-secreting cells upon *ex vivo* stimulation of splenocytes with GP33 peptide (Fig 5B and 5C). The data suggest that Bcl-3 had an inhibitory effect on acute CD8$^+$ T cell activation and proliferation in response to LCMV infection. A prior report also implicated Bcl-3 overexpression in slowing T cell proliferation in response to superantigen exposure [33]. In our *in vivo* LCMV infection study, further analysis of GP33$^+$ CD8$^+$ T cells for MPEC and SLEC subsets however revealed reduced proportions of SLECs and higher

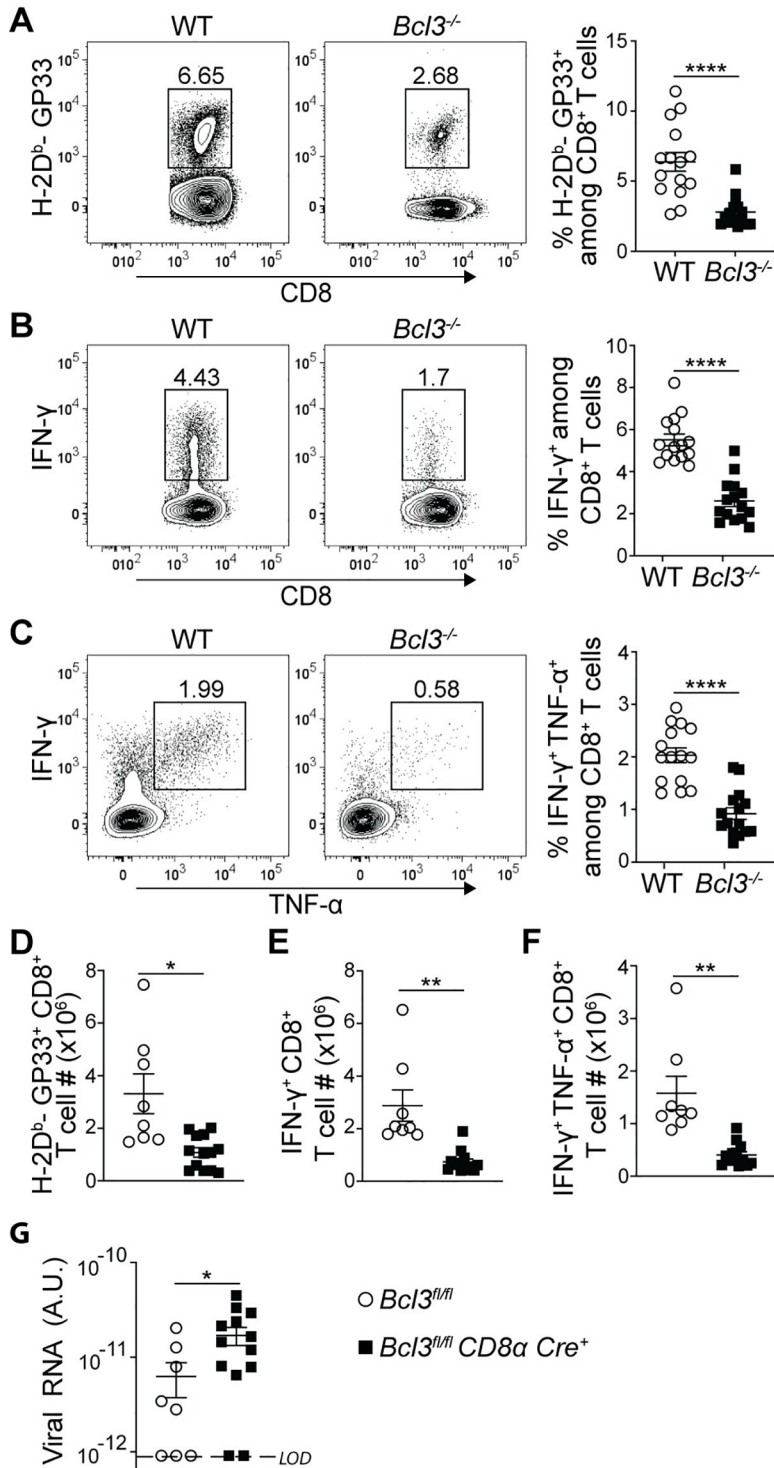

**Fig 4. Bcl-3 deficient CD8⁺ T cells are defective in their anti LCMV recall response.** (**A-C**) At least 8 weeks post LCMV Armstrong infection mixed bone marrow chimeras were challenged with LCMV clone 13. On day 5 post challenge, mouse spleens were analyzed for LCMV-specific CD8⁺ T cells. Plots are gated on WT or *Bcl3⁻/⁻* CD8⁺ T cells. Representative contour plots show proportions of H-2D^b-GP33 tetramer specific CD8⁺ T cells in spleen (**A**; left panels). Right graph summarizes data from three independent experiments with n = 15 mice for each CD8⁺ T cell genotype. Representative contour plots show proportions of IFN-γ (**B**; left panels) and IFN-γ + TNF-α (**C**; left panels) producing CD8⁺ T cells upon GP33 peptide stimulation *ex vivo*. Right graphs summarize data from three independent experiments with n = 15 mice for each CD8⁺ T cell genotype. (**D-F**) *Bcl3^{fl/fl}* and *Bcl3^{fl/fl} CD8α Cre* mice were infected

with LCMV Armstrong and at least 8 weeks later mice were challenged with LCMV clone 13. Spleens were analyzed at day 5 post challenge. Data are from 2 independent experiments with n = 8–13 mice for each genotype. (**D**) Absolute numbers of H-2D$^b$-GP33$^+$ CD8$^+$ T cells in spleen. Absolute numbers of IFN-γ (**E**) and IFN-γ + TNF-α (**F**) positive CD8$^+$ T cells upon GP33 peptide stimulation *ex vivo*. (**G**) Serum viral load after 4 days of LCMV clone 13 challenge of LCMV Armstrong infected mice. n = 8–13 mice for each genotype. LOD = Limit of detection. Error bars indicate SEM. *p<0.05, **p<0.01, ****p<0.0001.

proportions of MPECs in the presence of continuous Bcl-3 expression compared to controls (Fig 5D and 5E). In addition, the proportions of IFN-γ producers among GP33$^+$ CD8$^+$ T cells (comparing their absolute numbers) (Fig 5F) and TNF-α$^+$ cells among IFN-γ producers were also reduced in these Bcl-3 transgenic mice (Fig 5G). To confirm these findings, we additionally generated bone marrow chimeras in sub-lethally irradiated *Rag1*$^{-/-}$ mice using CD45.2$^+$ mice bearing the Bcl-3 transgene without *Cre* (WT-like, not expressing transgenic Bcl-3) and CD45.1$^+$ WT mice or from CD45.2$^+$ mice bearing the Bcl-3 transgene with *CD4-Cre* (expressing Bcl-3 in T cells) and CD45.1$^+$ WT mice. While the ratios of Bcl-3 transgene non-expressing/WT relative to Bcl-3 transgene-expressing/WT CD8$^+$ T cells were similar in uninfected chimeric mice, LCMV infection led to a reduction in the ratios of transgenic Bcl-3-expressing CD8$^+$ T cells relative to non-expressors at day 8 p.i. (S6A Fig). Among these CD8$^+$ T cells, those expressing the transgene exhibited notably lower ratios of LCMV antigen-specific cells (GP33$^+$ cells among CD8$^+$ T cells) in particular (S6B Fig), but at the same time, higher ratios of MPECs and lower ratios of SLECs among GP33$^+$ CD8$^+$ T cells (S6C Fig). Furthermore, ex vivo stimulation with GP33 yielded lower ratios of IFN-γ- and IFN-γ + TNF-α-producers in Bcl-3 transgene expressing CD8$^+$ T cells over controls (S6D Fig). Together, these data suggest that Bcl-3 is critical throughout T cell differentiation, wherein it first inhibits activation and

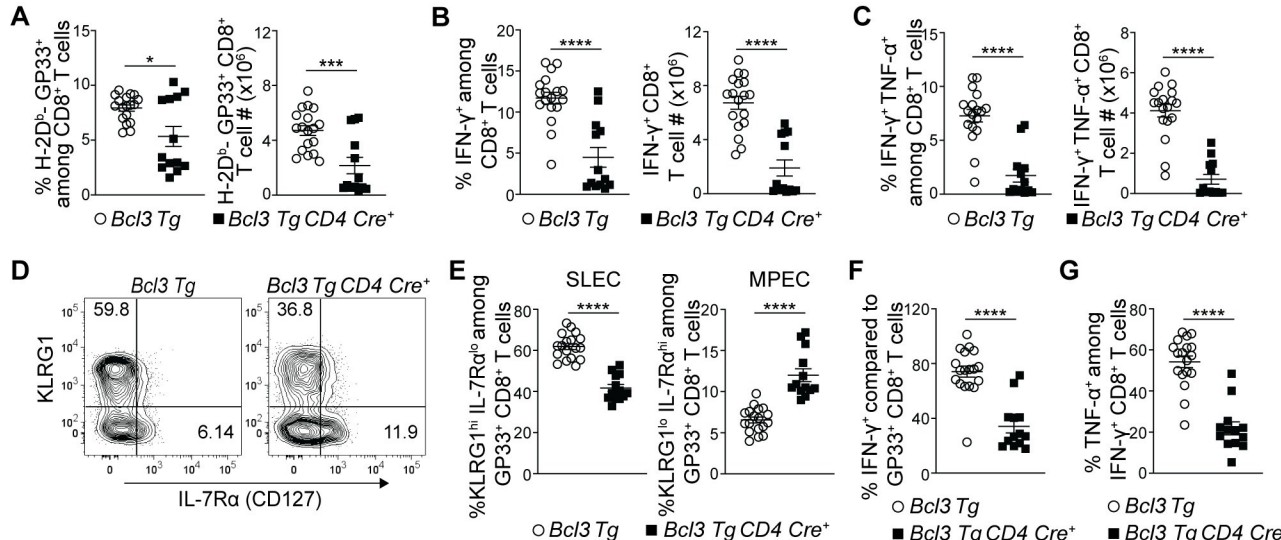

**Fig 5. Constitutive Bcl-3 expression restricts CD8$^+$ T cell terminal differentiation and supports formation of MPECs.** (**A-G**) *Bcl3 Tg CD4 Cre* and *Bcl3 Tg* (control) mice were infected with LCMV Armstrong, and spleens were analyzed 8 d p.i. (**A**) Graphs show proportions (left) and absolute numbers (right) of H-2D$^b$-GP33$^+$ CD8$^+$ T cells. Proportions (left panels) and absolute number (right panels) of IFN-γ (**B**) and IFN-γ + TNF-α (**C**) producing CD8$^+$ T cells upon GP33 peptide stimulation *ex vivo*. (**D**) Representative contour plots show proportions of SLEC and MPEC cells among H-D$^b$-GP33$^+$ CD8$^+$ T cells in spleen with surface markers as indicated. (**E**) Graphs summarize proportions of SLEC (left panel) and MPEC (right panel) subsets on H-D$^b$-GP33$^+$ CD8$^+$ T cells as shown in (**D**). (**F**) Relative percentages of IFN-γ producing cells among H-D$^b$-GP33$^+$ CD8$^+$ T cells. (**G**) Proportion of TNF-α producing cells among IFN-γ$^+$ CD8$^+$ T cells upon GP33 peptide stimulation *ex vivo*. Data are summarized from three independent experiments with n = 13–18 mice for each genotype. Error bars indicate SEM. *p<0.05, ***p<0.001, ****p<0.0001.

proliferation of CD8$^+$ T cells and possibly as a consequence, limits their terminal differentiation to support instead the formation of MPECs.

## Discussion

The acute effector response of CD8$^+$ T cells is key for efficient pathogen clearance, but it also shapes the quantity and quality of memory CD8$^+$ T cells, which are required for efficient immunity upon re-encounter with the pathogen at a later time. Thus, it is critical to understand how the complex heterogeneity of the acute CD8$^+$ T cell response is regulated at the molecular level in order to find the best strategies for efficient vaccinations and anti-tumor activities, both of which rely on robust memory. Limited previous studies have suggested a role for the oncogenic NF-κB regulator Bcl-3 in CD8$^+$ T cell differentiation, potentially aiding their survival *in vitro* upon activation [31] or boosting super-antigen-induced IFN-γ production upon secondary stimulation *ex vivo* [30]. However, the role of Bcl-3 in CD8$^+$ T cell differentiation and function during a pathogenic challenge *in vivo* has not been investigated, a process which begins with a complex, heterogeneous acute primary response, virus clearance, and ends with the formation of memory cells.

Here we explored the role of Bcl-3 in CD8$^+$ T cell differentiation and function with the LCMV infection model, a well-established model ideally suited for this purpose. We show that Bcl-3, acting cell-autonomously, is critical in the normal formation of memory precursor effector CD8$^+$ T cells (MPECs). We found at first glance that the magnitude of the acute primary response was not Bcl-3-dependent, since the overall proportion and absolute numbers of anti-LCMV-specific CD8$^+$ T cells that were generated were unaffected by the absence of Bcl-3 in these cells. However, deeper analysis involving single-cell RNA-seq as well as flow cytometry of differentiating CD8$^+$ T cell subpopulations revealed that the course of differentiation was fundamentally skewed. Lack of Bcl-3 pushed naïve CD8$^+$ T cells towards the terminal effector stage, while impeding their ability to become memory precursors. Terminal effector cells are destined to die during the contraction phase after the virus has been largely cleared, while memory precursors form longer lasting memory cells. This explains why we found notably fewer Bcl-3-deficient than Bcl-3-sufficient anti-LCMV CD8$^+$ T cells after the contraction phase, when terminal effectors have been mostly eliminated, while memory cells have persisted. The physiologic relevance of these findings was revealed in recall experiments, which showed that loss of Bcl-3 in CD8$^+$ T cells weaken their response to re-infection with LCMV and impaired clearance of the virus.

It is noteworthy that two distinct Bcl-3 knockout approaches employed here led to the same results. Mixed bone marrow chimeras containing both WT and KO CD8$^+$ T cells in the same mouse allow identification of purely CD8$^+$ T cell-autonomous effects of Bcl-3, independent of the surrounding milieu, whereas conditional knockouts of Bcl-3 specific to mature CD8$^+$ T cells allow identification of cell-intrinsic effects, i.e. loss of Bcl-3 in these cells only (not in any other cells); in the latter case, however, extrinsic feedback communication with other cells may occur to also affect the results. For example, CD8$^+$ T cells lacking Bcl-3 might change production of an extrinsically acting factor in a pathway involving other cells that then feeds back to the deficient CD8$^+$ T cells, thereby possibly altering their differentiation or function indirectly. In chimeric mice both WT and KO CD8$^+$ T cells are present and experience the identical milieu, so such indirect extrinsic effects are not detected. Since use of conditional CD8$^+$ T cell-specific knockouts yielded the same results as chimeric mice, our data suggest that indirect extrinsic effects did not occur or were negligible during both the primary and recall responses. It is possible of course that in different contexts indirect extrinsic effects might also contribute in addition to the cell-autonomous effects, including e.g. long-term chronic infections.

Bcl-3 is likely to modulate gene expression during a delayed-early phase in CD8$^+$ T cell activation/differentiation to alter the transcriptional profile in a way that suppresses terminal differentiation pathways. It may do so by partially modulating the expression of many of the NF-kB-regulated genes, rather than by changing expression of only a few select target genes. Bcl-3 RNA was induced early, though not immediately, upon activation in vitro, peaking by around 6 h. The Bcl-3 protein might of course have a more extended lifespan in the nucleus and could also be subject to functionally relevant post-translational modifications during that time. In any case, by day 8 p.i., the various subpopulations detected in KO and WT CD8$^+$ T cells appeared to have fairly similar transcriptional profiles. This suggests that once a CD8$^+$ T cell had chosen its path earlier during the infection, under the influence of Bcl-3, thereafter Bcl-3 no longer controlled gene expression, at least not the genes that could be detected in our scRNA-seq analysis. It will be of great interest to investigate gene expression during the first 24–48 hours after infection. The current acute natural infection model however does not generate enough cells to allow a full analysis at this early time. This issue might be approached in future experiments with transfer of sufficient numbers of antigen-specific T cells, however, such an approach creates an artificial situation when compared to the physiologic infection model employed here. Cell transfer experiments might also be useful to shed some light on the origin of the various effector subpopulations present at 8 d p.i. On the other hand, these subpopulations may simply represent different phases/activation stages of an effector cell at or near the height of the acute primary response, rather than reflecting truly distinct types of CD8$^+$ T cells.

The CD8$^+$ T cell effector response is central to clearing several bacterial, viral and fungal pathogens. CD8$^+$ T cells are also preferentially used in chimeric antigen receptor (CAR) T cell therapies to fight cancer. Since Bcl-3 deficiency led to an enhanced effector but impaired memory CD8$^+$ T cell response, it will be of great interest to investigate the role of Bcl-3 in CD8$^+$ T cells in other acute and persistent infection and tumor models. It is conceivable that manipulation of Bcl-3 regulation/expression in e.g. CAR CD8$^+$ T cells might allow for improved anti-cancer activities of these cells. A better understanding of the molecular targets of Bcl-3 as well as the regulation of Bcl-3 itself during CD8$^+$ T cell differentiation should further aid in these efforts.

## Materials and methods

### Ethics statement

All animal handling procedures and experiments were approved by the NIAID Animal Care and Use Committee (protocol LMI-23E).

### Mice

*Rag1*$^{-/-}$ (Taconic line 146) were purchased from Taconic Biosciences (Germantown, NY, USA). B6.SJL (CD45.1, JAX stock #002014), *Cd4-Cre* (JAX stock #17336) and *CD8α-Cre* (JAX stock #008766) mice were purchased from The Jackson Laboratory (Bar Harbor, ME, USA). *Bcl3 Tg* [38], *Bcl3*$^{-/-}$ and *Bcl3*$^{fl/fl}$ mice were described by us previously [39]. Anti-GP33 T cell receptor (P14) mice were generously provided by Dr. Ethan M. Shevach (NIAID/NIH) and crossed with *Bcl3*$^{-/-}$ mice. All mice bore the C57BL/6 background. To generate chimeric mice, WT (CD45.1) and *Bcl3*$^{-/-}$ *or Bcl3 Tg CD4 cre / control* (CD45.2) bone marrows were mixed in 1:1 ratio and transferred into sub-lethally irradiated (400 rads) *Rag1*$^{-/-}$ mice. Chimeric mice were infected with LCMV at least 6 weeks after reconstitution. All mice were housed in pathogen-free conditions in the NIAID animal facility.

## Viral infection

LCMV strains (Armstrong and Clone 13) were kind gifts from Dr. Ethan M. Shevach. Mice were injected with $2x\,10^5$ pfu of LCMV Armstrong intra-peritoneally for analyses of primary anti-LCMV T cell response and for generation of memory T cells. To check recall responses at least 8 weeks after prior infection with LCMV Armstrong, these mice were challenged via intravenous injection with $2 \times 10^6$ pfu of LCMV Clone 13. Viral titers of stocks were determined by plaque assay on Vero cell monolayer [40].

## Bcl-3 mRNA quantitation

CD8$^+$ T cells were purified from WT mice spleens by MACS with the CD8a$^+$ T cell isolation kit (Miltenyi Biotec, Auburn, CA, USA). Purified cells were kept at rest for 12 hours in T cell media (RPMI with 10% FBS, 10 mM HEPES, 1% non-essential amino acids, 1 mM sodium pyruvate, 100 units/ mL penicillin, 100 units/ mL streptomycin and 55 nM β-ME). After 12 h, cells were stimulated with plate bound CD3ε (145-2C11, 2 μg/mL) and soluble αCD28 (37.51, 1 μg/mL) (both from BioXcell, West Lebanon, NH, USA) in T cell media supplemented with murine IL-2 (10 ng/mL, PeproTech, Rocky Hill, NJ, USA). Cells were collected at various time points. For *in vivo* *Bcl3* kinetic analysis, CD8$^+$ T cells were purified from LCMV infected or naïve mice spleen by CD8a (Ly-2) microbeads (Miltenyi Biotec, Auburn, CA, USA) at various time points. RNA was prepared by RNeasy mini kit and cDNAs were synthesized using QuantiTect reverse transcription kit (both from Qiagen, Hilden, Germany). *Bcl3* levels were quantified relative to *Gapdh* by TaqMan assays on Quantstudio3 real time system (Applied Biosystems by Thermo Fisher Scientific, Carlsbad, CA, USA).

## Analysis of proliferation and apoptosis

CD8$^+$ T cells were purified from mouse spleen by MACS with the CD8a$^+$ T cell isolation kit (Miltenyi Biotec, Auburn, CA, USA) and stained with cell trace violet (CTV) proliferation dye eBioscience (Thermo Fisher Scientific, Eugene, OR, USA). $0.2\, x10^6$ cells were stimulated with plate bound anti-CD3ε (145-2C11, 2 μg/mL) and soluble αCD28 (37.51, 1 μg/mL) (both from BioXcell, West Lebanon, NH, USA) in T cell media supplemented with murine IL-2 (10 ng/mL, PeproTech, Rocky Hill, NJ, USA). Proliferation was measured by dilution of the proliferation dye and apoptosis was assessed with an Annexin V apoptosis detection kit with 7-AAD from BioLegend (San Diego, CA, USA). For *in vivo* proliferation comparison, CD8$^+$ T cells were purified from WT and *Bcl3* KO P14 mouse spleen by MACS with the CD8a$^+$ T cell isolation kit (Miltenyi Biotec, Auburn, CA, USA) and stained with cell trace violet (CTV) proliferation dye eBioscience (Thermo Fisher Scientific, Eugene, OR, USA). $1 \times 10^6$ cells were adoptively transferred to congenic recipient mice. The next day, mice were infected with LCMV Armstrong and mesenteric lymph nodes were analyzed 4 days after infection.

## Quantitation of serum IFN-γ levels

Day 5 post LCMV Armstrong infection, serum was collected from peripheral blood and stored in -80$^0$ C until analyzed. Serum samples were analyzed for IFN-γ with ELISA assay kit (R&D systems, Inc., Minneapolis, MN, USA) as per the manufacturer's instructions.

## Bcl-3 qPCR for knock down efficiency

Specific cell types were sorted from mouse spleen suspension. Genomic DNA were extracted from sorted cells with the help of Maxwell 16 Tissue DNA Purification Kit (Promega, Madison, WI, USA). qPCRs were done from extracted genomic DNA to determine

knockdown efficiency of *CD8a-Cre* expression by detecting Flox and knockout bands from Bcl-3 genomic region. For Flox band amplification: Fwd- CGTCCCCAGAGCCCGCAAC-CAC, Rev- GGGCCTCTCAACCTCTTTCCTA primer pair, while for knockout band: Fwd-GCGCCGCCCCGACTGAC, Rev- GGGCCTCTCAACCTCTTTCCTA primer pair were used.

## Flow cytometry

Single cell suspensions of spleens and LNs were stained with Live/Dead Aqua (Life Technologies Corporation, Eugene, OR, USA) along with surface antibodies as indicated as well as with GP33-MHC-I tetramers (NIH, NIAID tetramer core facility, Atlanta, GA, USA) at $4^0$ C for FACS analysis. For intracellular cytokine staining, splenocytes were stimulated with GP33 peptide (200 ng/mL, Peptide 2.0 Inc, Chantilly, VA, USA) for 4 hours in presence of protein transport inhibitor cocktail (eBioscience; Thermo Fisher Scientific, Carlsbad, CA, USA). Cells were stained for surface markers, fixed and permeabilized to stain for intracellular cytokines. Data were analyzed using FlowJo software (Becton Dickinson, Ashland, OR, USA). The following antibodies were used: CD8α (53–6.7), CD62L (MEL-14), Perforin (S16009A), Eomes (Dan11-mag), T-bet (ebio4B10), Granzyme B (NGZB) and CD127 (A7R34) from eBioscience (Thermo Fisher Scientific, Carlsbad, CA, USA), KLRG1 (2F1/KLRG1), IFN-γ (XMG1.2), TNF-α (MP6-XT22), CD45.1 (A20), CD45.2 (104) and PD1 (29F.1A12) from BioLegend (San Diego, CA, USA), CD44 (IM7) and CD3ε (145-2C11) from BD PharMingen (BD Biosciences, San Jose, CA, USA).

## Determination of serum virus titers

On day 4 post LCMV Clone 13 challenge, serum was isolated from peripheral blood of infected mice and stored at $-80^0$ C. Viral RNA was extracted with PureLink Viral RNA/DNA mini kit (Invitrogen; Thermo Fisher Scientific, Carlsbad, CA, USA) as per the manufacturer's instructions. Isolated RNA was reverse transcribed using QuantiTect reverse transcription kit (Qiagen, Hilden, Germany). The prepared cDNA was quantified by real-time PCR with PowerUp SYBR green master mix (Applied Biosystems; Thermo Fisher Scientific, Vilnius, Lithuania) on a QuantStudio 3 real time system (Applied Biosystems; Thermo Fisher Scientific, Carlsbad, CA, USA). For identification of LCMV transcripts, primers specific for the LCMV segment S envelope glycoprotein (GP) gene (Fwd-CATTCACCTGGACTTTGTCAGACTC, Rev-GCAACTGCTGTGTTCCCGAAAC) were used [41].

## Single-cell RNA sequencing

Splenic T cells were sorted from LCMV-infected chimeras, loaded onto a 10X Genomics Chromium platform to generate cDNAs carrying cell- and transcript-specific barcodes that were used to construct sequencing libraries using the Chromium Single Cell 3′ Library & Gel Bead Kit v2 according to the manufacturer instructions [37]. Libraries were sequenced on the Illumina NextSeq using paired-end 26x98bp in order to reach a sequencing saturation greater than 50%, resulting in 8,000–26,000 reads/cell. Data were analyzed as previously described [42]. Briefly, single-cell sequencing files were processed, and count matrices extracted using the Cell Ranger Single Cell Software Suite (v2.2.0). Further analyses were performed in R using the Seurat package (3.0) [43]. Data were pre-processed by removing genes expressed in fewer than 2 cells and excluding cells expressing fewer than 500 genes, or more than 10% mitochondrial genes. Reduction of data dimensionality was performed on the first 15 principal components (PC) calculated on the highly variable genes. Clustering was performed separately for each sample, and further analyses, including UMAP visualization, were performed after

merging datasets using the *MergeSeuratObjects* function. Clusters representing less than 1% of each population (which corresponds to the expected duplicate capture rate) were excluded from downstream analyses.

## Statistical analyses

Statistical significance is reported in figures and figure legends. Unpaired two-sided Student's *t*-test was performed to calculate statistical significance with GraphPad Prism software (version 8, San Diego, CA). If *F* test showed significant variance between groups, Welch's correction was applied.

## Supporting information

**S1 Fig. CD8$^+$ T cell stimulation induces *Bcl3* mRNA.** (**A**) *Bcl3* mRNA levels in purified CD8$^+$ T cells upon CD3 and CD28 stimulation. n = 4 mice. (**B**) *Bcl3* mRNA levels in bead-purified CD8$^+$ T cells from spleens of uninfected and LCMV Armstrong-infected WT mice (C57BL/6). n = 3 mice for each time point. (**C**) Ratio of CD4/CD8 T cells in blood of uninfected mixed bone marrow chimeric mice after at least 6 weeks of reconstitution. Data is representative of three independent experiments. n = 20 mice. (**D**) Knock down efficiency of *Bcl3* in *CD8α Cre* mice. Bcl3-Flox and knock-out bands were checked by qPCR from DNA of sorted cells of specific genotypes, as shown. (**E**) Spleens of uninfected *Bcl3$^{fl/fl}$* and *Bcl3$^{fl/fl}$ CD8α Cre$^+$* mice were analyzed for CD8$^+$ T cell subsets. n = 9–10 mice for each genotype. Error bars represents SEM. $^*$p<0.05, $^{**}$p<0.01, $^{***}$p<0.001, $^{****}$p<0.0001. ns = not significant. (TIF)

**S2 Fig. Bcl-3 restricts CD8$^+$ T cell differentiation.** (**A**) Experimental protocol: At least 6 weeks post reconstitution, mixed bone marrow chimeric mice were infected with LCMV Armstrong. At 8 d p.i., GP33 tetramer$^+$ CD8$^+$ WT or Bcl-3 KO cells were sorted and subjected to single cell RNA sequencing. (**B**) UMAP plots show the relative expression of the indicated genes in Bcl-3 KO and WT cells from the experiment shown in Fig 2. (**C**) Heatmap shows row-standardized expression of selected genes among clusters of Bcl-3 KO and WT cells in an independent experiment performed with another mixed bone marrow chimera and processed as described in Fig 2. For each genotype, percentages indicate the size of individual clusters relative to the total population. Cluster A', B$_1$'/ B$_2$', C', D$_1$'/D$_2$', E', F' and G' of the replicate set are roughly equivalent to clusters A, B, C, D, E, F and G of the first set in Fig 2, respectively (in this analysis the program distinguished more effector subsets). (TIF)

**S3 Fig. Bcl-3 supports formation of memory precursor CD8$^+$ T cells.** (**A** and **B**) GP33-epitope specific CD8$^+$ T cells from spleens of *Bcl3$^{fl/fl}$* and *Bcl3$^{fl/fl}$ CD8α Cre* mice were analyzed for SLEC and MPEC subsets on day 8 post-acute LCMV infection. n = 5–6 mice for each genotype. (**C**) *Bcl3* mRNA levels in various effector subsets of WT CD8$^+$ T cells specific for any of three LCMV epitopes (GP33, GP276, NP396), at day 8 post LCMV Armstrong infection. n = 4 mice. (**D-G**) GP33-epitope specific spleen CD8$^+$ T cells from *Bcl3$^{fl/fl}$* and *Bcl3$^{fl/fl}$ CD8α Cre* mice were analyzed for the indicated effector markers by flow cytometry at day 8 post-acute LCMV infection. n = 5–6 mice for each genotype. (**H**) Serum IFN-γ levels on day 5 post-acute LCMV infection. Data are cumulative from 3 independent experiments. Error bars represent SEM with n = 16–22 mice for each genotype. $^{**}$p<0.01, $^{***}$p<0.001, $^{****}$p<0.0001, ns = non-significant. (TIF)

**S4 Fig. Bcl-3 deficiency does not alter CD8$^+$ T cell proliferation and apoptotic potential.**
(**A-D**) 2 x 10$^5$ cell trace violet-stained purified CD8$^+$ T cells from $Bcl3^{-/-}$ and control mouse spleen were cultured for 72 hours in the presence of plate-bound anti-CD3ε (2 μg/mL) plus soluble anti-αCD28 (1 μg/mL) along with murine IL-2 (10 ng/mL). Data are averages of two independent experiments with n = 5 for each genotype. (**A**) Representative plot for cell division. (**B**) Percentages of nondividing and dividing cells. (**C**) Number of live cells by trypan blue staining. (**D**) Percentages of annexin V$^+$ (apoptotic) cells. (**E and F**) GP33-epitope specific CD8$^+$ T cells from spleens of $Bcl3^{fl/fl}$ and $Bcl3^{fl/fl}$ $CD8a$ $Cre$ mice were analyzed on day8 post-acute LCMV infection. Data are averages of two independent experiments with n = 10–11 mice for each genotype. (**E**) Percentages of annexin V$^+$ 7AAD$^-$ cells (apoptotic cells) among H-2D$^b$-GP33$^+$ CD8$^+$ T cells. (**F**) Percentages of annexin V$^+$ 7AAD$^+$ cells (dead cells) among H-2D$^b$-GP33$^+$ CD8$^+$ T cells. (**G and H**) 1x10$^6$ cell trace violet-stained purified WT or $Bcl3^{-/-}$ CD8$^+$ T cells from P14 transgenic mice were adoptively transferred to congenic recipient mice. On the next day, mice were infected with LCMV Armstrong and mesenteric lymph nodes were analyzed 4 days after infection. n = 4 mice in each group. (**G**) Representative plot for cell division. (**H**) Percentages of more dividing cells as gated in G. Error bars indicate SEM. ns = non-significant. (TIF)

**S5 Fig. Bcl-3 deficiency does not affect ability of CD8$^+$ T cells to form central memory T cells (T$_{CM}$).** (**A and B**) 8 weeks post LCMV Armstrong infection mixed bone marrow chimeras were analyzed for T$_{CM}$ and T$_{EM}$ subsets in spleen. (**A**) Representative contour plots show T$_{CM}$ (CD62L$^{hi}$ IL-7Rα$^{hi}$) and T$_{EM}$ (CD62L$^{lo}$ IL-7Rα$^{hi}$) subsets among H-2D$^b$-GP33$^+$ CD8$^+$ T cells. (**B**) Graph summarizes data from three independent experiments with error bars representing SEM. n = 15 mice for each CD8$^+$ T cell genotype. ns = not significant. (TIF)

**S6 Fig. Bcl-3 restricts terminal differentiation and supports MPEC formation.** (**A**) After at least 6 weeks of reconstitution, ratio of proportions of Bcl3 transgene non-expressing or expressing cells to WT cells among CD8$^+$ T cells in mixed bone marrow chimeric mice before infection (in blood) or at day 8 of LCMV Armstrong infection (spleen). Graph summarize data from two independent experiments with n = 10 mice each group. (**B-D**) After at least 6 weeks of reconstitution, mixed bone marrow chimeras were infected with LCMV Armstrong and spleens were analyzed 8 d p.i. Graphs summarize data from two independent experiments with n = 10 mice each group. (**B**) Ratio of proportions of GP33 tetramer$^+$ cells among CD8$^+$ T cells from chimeric mice containing a non-expressing Bcl-3 transgene or containing a Bcl-3 expressing transgene. (**C**) Ratio of proportions of KLRG1$^{hi}$ IL-7Rα$^{lo}$ cells (SLEC, left graph) and KLRG1$^{lo}$ IL-7Rα$^{hi}$ cells (MPEC, right graph) among GP33$^+$ CD8$^+$ T cells from chimeric mice containing a non-expressing Bcl-3 transgene or containing a Bcl-3 expressing transgene. (**D**) Ratio of proportions of IFN-γ$^+$ cells (left graph) and IFN-γ$^+$ TNF-α$^+$ cells (right graph) among CD8$^+$ T cells from chimeric mice containing a non-expressing Bcl-3 transgene or containing a Bcl-3 expressing transgene upon $ex$ $vivo$ stimulation of splenocytes with GP33 peptide. Error bars represents SEM. $^{**}$p<0.01, $^{****}$p<0.0001. ns = not significant. (TIF)

## Acknowledgments

We thank Ethan M. Shevach for LCMV strains; the NIH tetramer facility for providing tetramer reagents; NIAID flow cytometry core for cell sorting; Mike Kelly and Zachary Rae from CCR Single Cell Analysis Facility performed the capture and library preparation. Sequencing was performed by the CCR Sequencing Facility.

## Author Contributions

**Conceptualization:** Hemant Jaiswal, Ulrich Siebenlist.

**Data curation:** Hemant Jaiswal.

**Formal analysis:** Hemant Jaiswal, Thomas Ciucci.

**Funding acquisition:** Rémy Bosselut, Ulrich Siebenlist.

**Investigation:** Hemant Jaiswal, Hongshan Wang, Wanhu Tang, Estefania Claudio.

**Methodology:** Hemant Jaiswal, Thomas Ciucci.

**Project administration:** Hemant Jaiswal, Rémy Bosselut, Ulrich Siebenlist.

**Resources:** Hemant Jaiswal, Thomas Ciucci, Philip M. Murphy, Rémy Bosselut, Ulrich Siebenlist.

**Software:** Hemant Jaiswal, Thomas Ciucci.

**Supervision:** Ulrich Siebenlist.

**Writing – original draft:** Hemant Jaiswal, Thomas Ciucci, Rémy Bosselut, Ulrich Siebenlist.

**Writing – review & editing:** Hemant Jaiswal, Thomas Ciucci, Philip M. Murphy, Rémy Bosselut, Ulrich Siebenlist.

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
