## [Decision Letter · Decision Letter 0]

2 Oct 2020

Dear Dr. Siebenlist,

Thank you very much for submitting your manuscript "The NF-κB regulator Bcl-3 restricts terminal differentiation and promotes memory cell formation of CD8+ T cell in viral infection" for consideration at PLOS Pathogens. As with all papers reviewed by the journal, your manuscript was reviewed by members of the editorial board and bytwo independent reviewers. In light of the reviews (below this email), we would like to invite the resubmission of a significantly-revised version that takes into account the reviewers' comments.

We specifically request you to provide data for the following comments.

a) Show whether Bcl3 is expressed on different CD8 T cell subsets during LCMV infection

b) Demonstrate if there is differential proliferation and or apoptosis of Wt and Bcl3-/- CD8 T cells

c) (Optional) show the expression of perforin, gran B, Tbet, Eomes and other effector molecules of CD8 T cells in wt and Bsl3-/- mice.

d) Provide control gates for Fig2 C/D FACS plots

In addition, we request authors to provide explanation for the other comments raised by the reviewers.

We cannot make any decision about publication until we have seen the revised manuscript and your response to the reviewers' comments. Your revised manuscript is also likely to be sent to reviewers for further evaluation.

Sincerely,

Rudra Channappanavar, DVM, Ph.D.

Guest Editor

PLOS Pathogens

Volker Thiel

Section Editor

PLOS Pathogens

Kasturi Haldar

Editor-in-Chief

PLOS Pathogens

orcid.org/0000-0001-5065-158X

Michael Malim

Editor-in-Chief

PLOS Pathogens

orcid.org/0000-0002-7699-2064

Reviewer's Responses to Questions

**Part I - Summary**

Reviewer #1: In this manuscript, authors have characterized the role of Bcl-3 in terminal differentiation of memory CD8+ T cells. Using acute LCMV model, investigators show that BCL3 is critical for maintaining balance of terminal effector/memory cell balance during CD8+ T cell differentiation. This study lacks mechanistic details for observed differences in SLECs and MPECs post LCMV infection and recall responses. Thus, descriptive phenotypic changes in Knock-out mice in the absence of mechanistic details diminishes the importance of these findings. Nevertheless, overall findings are interesting and various conclusions are well supported by data generated using WT/Bcl-3 KO bone-marrow chimera and CD8-specific conditional Bcl-3 knock-out mice.

Reviewer #2: This study is very interesting as it show the role of Bcl3 in generation of cd8 memory t cells. The study is well designed and uses various high end techniques and tools to make a conclusion.

**Part II – Major Issues: Key Experiments Required for Acceptance**

Reviewer #1: 1. Authors show transient expression of Bcl-3 in CD8+ T cells post CD3/CD28 stimulation. Although this data indicates the transient induction of Bcl-3 post TCR stimulation, similar expression kinetics data for Bcl-3 in naïve, effector/memory CD8+ T cells post LCMV infection is critical as a premise to study the role of Bcl-3 in in vivo differentiation of CD8+ T cells post LCMV infection.

2. Mixed bone-marrow chimera experiments do not show overall differences in LCMV-specific CD8+ T cell responses after acute infection. Further, mixed chimera experiments do not rule out the role for Bcl-3 in other myeloid cells particularly APCs in regulation of effector versus memory T cell differentiation. Although authors have used conditional knock-out mice for CD8+ T cells to demonstrate cell-intrinsic role of Bcl-3 in efficient formation of memory CD8+ T cells, the choice of cre-strain indicate that there could be developmental defects in CD8+ T cells this conditional knock-out mice.

3. There is no clear mechanistic explanation for increased terminal effector cells over MPECs in Bcl-3-/- mice studies. Is this because of increased proliferation or increased survival/reduced apoptosis? Also, mixed bone marrow-chimera experiment show differential reconstitution of CD4/CD8 T cells in WT and KO cells. This data points initial differences in WT and KO cells could drive differential CD8+ T cell responses post LCMV challenge.

4. Data represents multiple experiments. Did authors pool data from different experiments? If yes, was there a similar statistical difference for each individual experiment?

Reviewer #2: Jaiswal et al describes the cell autonomous role for Bcl-3 in CD8+T cell differentiation during the response to lymphocytic choriomeningitis virus infection using single-cell RNA-seq and flow cytometric analysis of virus specific Bcl3-/- CD8+T cells.

Although this study identify the role of Bcl3 in generation of CD8 memory response, it is not clear as to why bcl3 regulate the generation of MPECs. Whether bsl3 somehow controlling activation induced cell death and therefore as a result a reduced frequency of MPECs were found in bsl3 deficient mice. Authors must address this. In addition, authors should show whether Bsl3-/- cd8 t cells proliferate less both in vitro and in vivo. Authors should also show the expression of perforin, gran B, Tbet, Eomes and other effector molecules of CD8 T cells in wt and Bsl3-/- mice.

**Part III – Minor Issues: Editorial and Data Presentation Modifications**

Reviewer #1: (No Response)

Reviewer #2: (No Response)

Reviewer #1: No

Reviewer #2: No
---

## [Editor Report · Decision Letter 1]

21 Dec 2020

Dear Dr. Jaiswal,

We are pleased to inform you that your manuscript 'The NF-κB regulator Bcl-3 restricts terminal differentiation and promotes memory cell formation of CD8+ T cells during viral infection' has been provisionally accepted for publication in PLOS Pathogens.

Best regards,

Rudra Channappanavar, DVM, Ph.D.

Guest Editor

PLOS Pathogens

Volker Thiel

Section Editor

PLOS Pathogens

Kasturi Haldar

Editor-in-Chief

PLOS Pathogens

orcid.org/0000-0001-5065-158X

Michael Malim

Editor-in-Chief

PLOS Pathogens

orcid.org/0000-0002-7699-2064
---

## [Editor Report · Acceptance letter]

22 Jan 2021

Dear Dr. Jaiswal,

We are delighted to inform you that your manuscript, "The NF-κB regulator Bcl-3 restricts terminal differentiation and promotes memory cell formation of CD8+ T cells during viral infection," has been formally accepted for publication in PLOS Pathogens.

Best regards,

Kasturi Haldar

Editor-in-Chief

PLOS Pathogens

orcid.org/0000-0001-5065-158X

Michael Malim

Editor-in-Chief

PLOS Pathogens

orcid.org/0000-0002-7699-2064